# Internal conflict and prejudice-regulation: Emotional ambivalence buffers against defensive responding to implicit bias feedback

**Naomi B. Rothman** [1]*, **Joseph A. Vitriol**[2], **Gordon B. Moskowitz**[3]

**1** Department of Management, Lehigh University, Bethlehem, PA, United States of America, **2** Department of Political Science, Stony Brook University, Stony Brook, NY, United States of America, **3** Department of Psychology, Lehigh University, Bethlehem, PA, United States of America

* nbr211@lehigh.edu

**Data Availability Statement:** All relevant data files are available from the OSF database (https://osf.io/ yre38/?view_only= e6ec0f6759a44bc9ab6c982eaacacf99).

## Abstract

Becoming aware of bias is essential for prejudice-regulation. However, attempts to make people aware of bias through feedback often elicits defensive reactions that undermine mitigation efforts. In the present article, we introduce state emotional ambivalence—the simultaneous experience of positive and negative emotions "in the present moment"–as a buffer against defensive responding to implicit bias feedback. Two studies (N = 507) demonstrate that implicit bias feedback (vs. no feedback) increases defensiveness (rating the test as less valid, credible, and objective). However, high (vs. low) state emotional ambivalence, which was independent of bias feedback, attenuates this relationship between bias feedback and defensiveness, accounting for a larger share of the variance than negative emotions alone. In turn, this reduced defensiveness among individuals high (vs. low) in emotional ambivalence was associated with increased awareness of bias in the self and others. Results suggest that state emotional ambivalence is associated with increased bias awareness by creating a mindset in which individuals are less defensive to potentially threatening information about their *own* implicit racial bias. These results have important implications for research on stereotyping and prejudice, emotional ambivalence and psychological conflict, and defensiveness.

## Introduction

White Americans are divided on the existence and consequences of racial prejudice, especially its more subtle or unconscious manifestations [1,2]. Many believe that prejudice disadvantages minority groups [3], and are motivated to be egalitarian and fair in their own judgment [4]. However, others disagree and believe that racial prejudice and discrimination are problems of the past [1,5], that society is fair and just [6]), and that efforts to remediate racial disparities and reduce prejudice are misguided and counter-productive [2,7].

Since most people often do not recognize prejudice in society or themselves [8], and even tend to believe they are more objective and less prejudiced than others [9], many scholars and

**Funding:** The author(s) received no specific funding for this work.

**Competing interests:** The authors have declared that no competing interests exist.

practitioners have proposed that directly demonstrating a person's bias is an important first step towards increasing awareness of bias in the self. Without such a realization one would be relatively unmotivated to change behavior and engage in prejudice-regulation [10–12]. Unfortunately, confronting people with evidence that they are prejudiced can also incite defensive responding, such as trivializing, derogating, or avoiding the feedback altogether [13–15].

Research suggests that to successfully initiate prejudice regulation in others, bias awareness must be raised, and to do this requires a non-defensive reaction to discussions about bias and feedback about personal biases [16]. Previous research on reducing defensiveness has focused solely on how to mitigate the threat posed by bias feedback through the wording of the feedback [17]. However, aside from shaping the nature of the message, another approach to reducing defensiveness is to focus on preparing the message recipient for potentially threatening information. This is the approach we take through examining the state of emotional ambivalence, experienced in the present moment.

In this article, we integrate previous research on defensive responding and prejudice-regulation with research on ambivalence and psychological conflict to propose that the state of emotional ambivalence, the simultaneous experience of positive and negative emotions in the present moment, can make people less defensive to feedback about implicit racial bias. We theorize that emotional ambivalence plays this mitigating role by making people more cognitively flexible and thus less defensive in their processing of the feedback. For people high in emotional ambivalence, this attenuated defensiveness, in turn, makes them more aware of bias in the self and others.

## Differences between attitudinal ambivalence and emotional ambivalence

Emotional ambivalence can be differentiated from traditional approaches to studying attitude ambivalence. Traditional approaches to the study of *attitude* ambivalence build on the dissonance literature. Dissonance is caused by a threatening consistency violation (e.g., threat to the integrity of the self-system [18]),—usually the result of behavioral or attitudinal commitment to a cognition that is in direct conflict with a pre-existing attitude. The threat is greatest when a self-relevant attitude or belief is challenged by an inconsistent (dissonant) action, belief, or attitude. For instance, dissonance may be triggered when a self-relevant attitude is directly challenged by some new information, such as bias feedback. The conflict creates threat, and people become motivated to remove the threat in a variety of ways, with defensiveness being one predominant strategy. Similarly, traditional approaches to the study of attitude ambivalence assume that ambivalence is a threatening consistency violation, is usually experienced as unpleasant, and motivates desires to reduce the ambivalence or the negative affect it produces in a variety of ways.

However, not all attitude ambivalence is experienced as threat. There is evidence that attitude ambivalence does not always produce negative affect or threat. Attitudinal ambivalence, for instance, can be desirable when an issue is controversial [19], is sometimes negatively related to physiological arousal [20], and is even cultivated by people in the face of uncertainty that they can obtain a desirable outcome [21]. In fact, ambivalent attitude holders are thought to only experience more arousal than individuals with univalent attitudes when they need to commit to one side of the attitude object [22].

This perspective is consistent with how we conceptualize the state experience of emotional ambivalence. As we define it, the state experience of emotional ambivalence–an experience of positive and negative emotions "in the present moment"–is non-threatening because it is not about a specific triggering event (e.g., feedback about the self), nor is it a response to it. In short, it is not a reaction against a threat to the integrity of the self-system, which one must

defend. It is simply the existence of positive and negative emotions at the same time, in the present moment. Further, emotional ambivalence is expected to be non-threatening in our research because people do not have to make an unequivocal stance and can remain noncommittal (there is no behavioral commitment). In our paradigm, therefore, emotional ambivalence is not expected to be self-threatening for these reasons [23] and is thus not expected to produce defensiveness.

Rather, our expectation is that when this non-threatening state experience of emotional ambivalence is high, it leads to reduced defensiveness in cognitive processing about bias feedback. We draw on response amplification theory [24,25] which suggests that people who are ambivalent are more influenced by the context than people who are not ambivalent. For instance, research has demonstrated that compared to their unambivalent counterparts, ambivalent attitude holders are prone to exhibiting more extreme responses–behavioral intentions—to a stigmatized individual in the direction of the positive or negative information provided about that individual. In contrast, these situational cues do not impact unambivalent participants who are less open to considering information from sources other than their prior attitudes [25]. By extension, we predict that emotional ambivalence will also make people more influenced by contextual cues–that is, less defensive to information provided by the experimenter. In our research, that information is feedback about their implicit race bias.

Some suggestive evidence for this prediction comes from research showing that individuals who are high in attitude ambivalence are more open to persuasion. For example, they can be persuaded by individuals from both their own university and another university, whereas low ambivalent individuals are less open to persuasion, only demonstrating attitude change when the source of the message is the student at their own university [26]. Our work aims to advance this scholarship by demonstrating that the state experience of emotional ambivalence can make people less defensive and more influenced by feedback from others, even when that feedback is potentially self-threatening.

## A more flexible and broader processing style

Evidence from across numerous different literatures studying internal conflict suggest a reason why emotional ambivalence should make people less defensive and more influenced by even negative information about the self [27,28]. Internal conflicts that are not threatening, such as emotional ambivalence [29], mind-body dissonance [30], nonconscious goal conflicts [31], and metaphorical conflict [32] occur when contradictory alternatives are simultaneously present. This research has shown that creating internal conflict activates a broader general reasoning process that, once accessible, can be applied to subsequent judgments [29,32]. Presuming the internal conflict is not a threat.

Specifically, internal conflict broadens general reasoning by increasing the individuals' scope of attention, and also motivates a balanced consideration of multiple different perspectives. For instance, non-conscious goal conflicts increase a broader and more balanced consideration of relevant information before making choices; specifically, participants who experienced non-conscious goal conflict searched for more information by electing to see a larger number of boxes on an information display board [33] and also sought and considered both confirmatory and disconfirmatory information in a trait hypothesis testing task [31]. Further, emotional conflict increases motivation to consider both positive and negative information about others before making decisions; specifically, participants who experienced emotional ambivalence in the present moment were motivated to consider both positive and negative feedback about a potential job candidate, in comparison to happy participants who were more motivated to seek positive than negative feedback [29]. In another study,

participants who experienced incidental emotional ambivalence through a recall exercise were more likely to seek, weigh, and incorporate alternative perspectives—measured by others' advice—while making numerical estimations, relative to participants who experienced happiness or sadness through a recall exercise [29]. As a result, it has been suggested that emotional ambivalence increases cognitive flexibility [28].

Sassenberg et al. [34] provide a review of research on the flexibility mindset, which is defined as an "activated cognitive strategy that leads to more divergent thinking, the use of broad cognitive categories, and the switching between categories" (p. 3). Such a mindset is triggered by situations that call for creativity and where internal conflict is present. As others before [28], they discuss emotional ambivalence as one type of conflict that should trigger this divergent, flexible and broad style of processing that creates greater possibilities for both persuasion and self-regulation. In the present research, we build on this prior work [28] and theorize that emotional ambivalence creates a cognitively flexible mind-set that broadens the scope of attention and will motivate individuals to engage in a balanced consideration of different perspectives. We predict that this cognitively flexible mindset will extend to information about the existence of one's *own* implicit racial bias. This will render emotionally ambivalent individuals less defensive to self-relevant and potentially self-threatening information, attenuating defensive responses to personal implicit bias feedback.

### The relationship of emotional ambivalence and prejudice regulation

Cognitive flexibility is central to the broader processing observed in research on creativity [35–37], persuasion [38], deliberation among alternative goal pursuits [39], and most importantly for the current purpose, stereotype change [40]. As a result, there is reason to believe emotional ambivalence should be important for not only attenuating defensive responses to bias feedback, but, in turn, to increasing bias awareness.

Prejudice regulation, specifically a decrease in the experience and expression of prejudice, requires people to be motivated to alter their responding. To do so often first requires awareness that responding is biased and in need of alteration. This is interfered by defensive processing during discussions about bias [8,17,41]. Many people respond defensively to implicit bias feedback [8,15,14].

But, we propose that reduced defensiveness to bias feedback from increased emotional ambivalence will, in turn, increase concern about and awareness of one's implicit racial bias. Because people who are high in bias awareness are attuned to the possibility that they exhibit subtle biases [8], this awareness should be higher among individuals who are less (vs. more) defensive. Thus, we expected a conditional indirect effect of implicit bias feedback (vs. no feedback) on bias awareness through reduced defensive responding, with individuals high (vs. low) in emotional ambivalence showing reduced defensive responding and, in turn, increased bias awareness. Examining reactions to implicit racial bias feedback is a fruitful domain for testing whether emotional ambivalence reduces defensiveness to threatening information about the self, and thus greater awareness of bias. To our knowledge, emotional ambivalence has not been examined in the prejudice-regulation domain.

## Current research

### Overview and design

Across two independent samples, we investigate the hypothesis that the effect of implicit bias feedback (vs. no feedback) on defensive responding will be reduced among individuals who are high (vs. low) in emotional ambivalence (Hypothesis 1). We also examine whether defensive responding will mediate the relationship between the interaction between bias feedback

and emotional ambivalence on bias awareness. We expect that, among people high (vs. low) in emotional ambivalence, implicit bias feedback (vs. no feedback) will indirectly covary with increased bias awareness through reduced defensive responding (Hypothesis 2).

We test these hypotheses using a meta-analysis of observations from two independent samples. Both samples employed a single independent variable design (Implicit Racial Bias Feedback vs. No Feedback). In Sample 1, an additional experimental condition was run concomitantly, but was designed to test hypotheses different from what is addressed here. Information about this condition is available in the Supplemental Materials. Additional measures were assessed in Sample 2 for research questions not addressed in the current study. These are also available in the Supplemental Materials. Below, we describe the characteristics of each sample separately. We then present meta-analytic estimates across the two samples for tests of our hypotheses. The Supplemental Materials provide a complete description of all of the measures, analyses, and results for each sample, separately, including measures not included in the current analyses. All other measures, manipulations, and exclusions are fully reported.

## Materials and methods

### Participants

The University of Minnesota Institutional Review Board indicated approval of this research via written consent. Participants for both samples were recruited from Amazon MTurk. Although these samples are not a representative, random sample of the American public, Mturk samples are older and more diverse than typical samples of university students, and more nationally representative than typical internet samples [42]. By utilizing Mturk, we were able to obtain a large, non-random sample of White Americans with sufficient variability on demographic characteristics and, more importantly, the constructs of interest (see [43], on the usefulness of Mturk for psychological research).

**Sample 1** included 268 U.S. citizens (64.93% females, 35.07% males; age $M$ = 36.19, $SD$ = 12.23; 44.78% report a family income greater than 50K and 46.27% have earned at least a Bachelor's degree). As planned, 30 non-White participants were excluded from analyses, as our focus is primarily on White Americans, leaving us with a final sample of 238 White U.S. Citizens. With the current sample size, to observe an interaction between feedback condition and emotional ambivalence, we estimated that we had 34% power to detect a Cohen's $d$ of 0.2 and 97% power to detect a Cohen's $d$ of 0.5, and 99% power to detect Cohen's $d$ larger than .5.

**Sample 2** included 317 U.S. citizens (59.9% females, 40.10% males; age $M$ = 37.40, $SD$ = 13.11; 49.10% report a family income greater than 50K and 54.7% have earned at least a Bachelor's degree). As planned, 48 non-White participants were excluded from analyses, as our focus is primarily on White Americans, leaving us with a final sample of 269 White U.S. Citizens. With the current sample size, to observe an interaction between feedback condition and emotional ambivalence, we estimated that we had 37% power to detect a Cohen's $d$ of 0.2 and 98% power to detect a Cohen's $d$ of 0.5, and 99% power to detect Cohen's $d$ larger than .5.

### Procedure

Participants were recruited for a study of "Attitudes About People". The study advertised that it was primarily looking to recruit White U.S. citizens. The name of the study is intended to increase the expectation that one's beliefs and attitudes about other people would be directly measured.

Participants first viewed a consent form for the study, and were then randomly assigned to the bias feedback or no feedback (i.e., control) condition. Participants then proceeded to

complete an IAT that they were told would measure "unconscious racial attitudes". The test presented them with pictures of men they needed to categorize according to race, and words they needed to categorize as good or bad, with accuracy and speed supposedly being measured for the purpose of yielding a "bias score" that would later be reported to them [44]. This test was not actually used to provide the feedback, as it was merely a cover story to provide participants with a basis for feedback that was, in reality, randomly manipulated. That is, this test was used as a false-feedback paradigm [17,45], in which participants were randomly assigned to receive bias feedback (vs. no feedback). The actual validity of the IAT as a measure is irrelevant to our purpose.

Our main objectives were to measure defensive reactions following exposure to the bias feedback. In these studies, the bias feedback is always a *deception* by experimental design–the feedback is not accurate and our goal is to have participants believe it is accurate, in order to impactfully distribute feedback about bias. The validity of utilizing the IAT as a paradigm for manipulating beliefs about personal attitudes was first demonstrated by Vitriol et al.5 and has since been used in research studying reactions to implicit bias feedback [17]. Participants complete the IAT, and receive false feedback about their performance (though we use their actual scores as control variables). The language and stimuli used for the feedback is available in the Supplemental Materials.

After receiving bias feedback, all participants complete measures of affect, defensive responding, and bias awareness. All of these measures are described below. Finally, participants completed measures of demographics and were fully debriefed.

## Measures

Means (SD), alphas, and intercorrelations of all measures are available in Table 1. The exact language used in the instructions, question stems, and items for each experiment, are available in the Supplemental Materials. All continuous variables were rescaled to run from 0–1 for easier comparison and estimation of effect sizes.

**Implicit attitudes.** The IAT consists of two critical blocks: in one block the labels "White People" and "Good" share the same response key, and "Black People" and "Bad" share another response key. A trial involves a stimulus appearing at the center of the screen, which corresponds to one of the four labels, and the correct response key must be made before moving onto the next trial. In the other critical block, the instruction is reversed and the labels "White People" and "Bad" share the same response key and "Black people" and "Good" share the same response key. If participants have faster reaction times to the first block relative to the second block, this indicates a pro-White/anti-Black. The magnitude of this difference is reflected in a participant's *D*-score (see [46]).

**Table 1. Mean, SD, and correlations between all continuous variables used.**

| Variables | *M* | *SD* | 1 | 2 | 3 | 4 |
|---|---|---|---|---|---|---|
| 1. Race IAT D-Scores | .63 | .07 | – | | | |
| 2. Emotional Ambivalence | .31 | .19 | .04 | – | | |
| 3. Defensive Responding | .59 | .28 | -.03 | -.12** | – | |
| 4. Bias Awareness | .48 | .18 | .00 | .37** | -.45 ** | – |

*Note.*
†*p* < .10
*\*p* < .05
*\*\*p* < .01.

**Affect.** Following the procedure described by prior research on prejudice-regulation [47], participants reported the extent to which affect adjectives characterized their emotional state (1 = does not apply at all, 7 = applies very much). Six items were used to measure positive affect (e.g., "optimistic"; Sample 1 Cronbach's alpha = .92, Sample 2 Cronbach's alpha = .94) and 7 items were used to measure negative affect (e.g., "guilty"; Sample 1 Cronbach's alpha = .95, Sample 2 Cronbach's alpha = .95). Specifically, participants reported to what extent each statement was representative of them in the present moment. To assess ambivalence, we calculated an ambivalence score to reflect the balance between positive and negative affect [48], assessing both similarity and extremity in the coexisting positive and negative components. For instance, an individual who reports the strongest positive and negative feelings would be considered highly ambivalent, whereas an individual with high positive and low negative (or vice versa) feelings would have low ambivalence. The Griffins formula, is one of the most commonly used calculations of ambivalence in the literature, is ((P + N)/2 - |P–N|), where P = positive and N = negative emotions. Higher values represent higher levels of emotional ambivalence.

**Defensive responding.** Following the procedure used in other research examining reactions to implicit bias feedback [17], participants reported their belief in the validity, credibility, and objectivity of the IAT across 4 items using a 7-point scale (1 = Not at all, 7 = Extremely"). These items include 1) "In your opinion, how credible is this test?", 2) "In your opinion, how objective is this test?", 3) "In your opinion, how valid are the results of this test?, 4) "In your opinion, how useful is this test for understanding people's racial attitudes?" These items were reverse-coded, such that higher values represent lower perceptions of the credibility and validity of IAT, and thus higher levels of defensiveness (Sample 1 Cronbach's alpha = .90, Sample 2 Cronbach's alpha = .94).

**Bias awareness.** Participants reported the extent to which they perceive themselves as biased. 13-items measured participants' recognition of their own implicit racial bias and its social consequence. On a 7-point scale, participants responded to such items as, "How likely is it that your unconscious beliefs are unfavorable toward racial minorities?", "Do you believe that your unconscious racial attitudes influence your behavior towards racial minorities in an unfair way?", and "How likely is it that unconscious racial attitudes biases people's judgments and behavior towards racial minorities?" Higher values represent increased bias awareness (Sample 1 Cronbach's alpha = .89, Sample 2 Cronbach's alpha = .90).

**Demographics.** Participants reported their age, gender, race, family income, and level of education.

## Results

The data and data syntax for the analysis reported below are available at: https://osf.io/yre38/?view_only=e6ec0f6759a44bc9ab6c982eaacacf99.

To test our first hypothesis, we first conducted a meta-analysis across the two samples, in which we estimated a multilevel model with maximum likelihood estimation and sample submitted as a random-intercept term. A dummy-coded variable was created to represent condition assignment, with the bias feedback condition coded as "1" and the no feedback treated as the reference group (coded as "0). Interaction terms were constructed between emotional ambivalence and this dummy-coded variable. Simple slope analyses for significant interactions were computed at one standard deviation above and below the mean of the moderator, following the procedures recommended by Aiken and West [49].

The results of the random-intercept model support Hypothesis 1. We obtained a significant interaction between bias feedback (vs. no feedback) and emotional ambivalence (Z = -3.26, b =

Effect of Bias Feedback on Defensiveness by Emotional Ambivalence

**Fig 1. Effect of bias feedback on defensiveness by emotional ambivalence.** (A) Study 1: The effect of feedback condition on defensive responding as a function of emotional ambivalence (B) Study 2: The effect of feedback condition on defensive responding as a function of emotional ambivalence (C) Meta-Analytic Analysis: The effect of feedback condition on defensive responding as a function of emotional ambivalence.

-.38, $SE = .12$, $(95\% \ CI = -.61, -.15)$, $p = .001$). Bias feedback (vs. no feedback) was found to increase defensive responding among individuals 1 SD below the mean of emotional ambivalence ($Z = 6.21$, $b = .39$, $SE = .06$, $(95\% \ CI = .27, .51)$, $p < .001$), but this effect was attenuated at 1 SD above the mean of ambivalence ($Z = 9.28$, $b = .25$, $SE = .03$, $(95\% \ CI = .19, .30)$, $p < .001$). Given that all original variables were rescaled to run from 0–1, substantively these estimates indicate that for participants with low levels of emotional ambivalence, bias feedback (vs. no feedback) led to approximately a 39% increase in defensive responding; for participants with relatively high levels of emotional ambivalence, bias feedback (vs. no feedback) led to approximately a 25% increase in defensive responding. The effects of feedback condition on defensive responding as a function of emotional ambivalence is provided in Fig 1, separately for each sample and for the meta-analytic estimate. These results are robust to the inclusion of Race IAT D-scores and univariate affect (positive and negative) as control variables, suggesting that the interaction between emotional ambivalence and feedback remain significant when accounting for univalent positive and negative emotion. No significant main effect of feedback condition on emotional ambivalence was observed.

We also conducted an analysis in which we estimated the simultaneous effect of three interaction terms—feedback condition and positive emotion, negative emotion, or ambivalent emotion—on defensive responding, again using a multilevel model with maximum likelihood estimation and sample submitted as a random-intercept term. Doing so renders all interaction terms statistically non-significant ($ps > .05$), which is to be expected, given that ambivalence is computed using positive and negative emotion. Nonetheless, we estimated two models in which we regressed defensive responding on positive, negative, and ambivalent emotion, separately for participants in the Bias Feedback and No Feedback condition. Again, we used multilevel models with maximum likelihood estimation and sample submitted as a random-intercept term. For participants in the No Feedback condition, positive emotion significantly predicted defensive responding ($b = -.27$, $SE = .07$, $(95\% \ CI = -.41, -.13)$, $p < .001$), but neither negative emotion ($b = -.05$, $SE = .12$, $(95\% \ CI = -.29, .19)$, $p = .69$) nor ambivalent emotion ($b = -.08$, $SE = .14$, $(95\% \ CI = -.35, .19)$, $p = .54$) was a significant predictor. In contrast, for

participants in the Bias Feedback condition, positive emotion did not significantly predict defensive responding ($b$ = -.10, $SE$ = .06, *(95% CI = -.22, .01)*, $p$ = .08). However, both negative emotion ($b$ = -.20, $SE$ = .08, *(95% CI = -.35, -.04)*, $p$ = .013) and ambivalent emotion ($b$ = -.24, $SE$ = .10, *(95% CI = -.44, -.04)*, $p$ = .02) were significant predictors.

More importantly, because all variables were rescaled to run 0–1, we are able to compare the strength of the relationship between defensive responding and negative or ambivalent emotion in the Bias Feedback condition. Inspection of the unstandardized coefficients indicates that moving from the lowest to the highest levels of negative emotion corresponded with approximately 20% reduction in defensive responding (while controlling for ambivalent emotion and positive emotion), whereas moving from the lowest to the highest levels of ambivalence corresponded with approximately 24% reduction in defensive responding (while controlling for negative and positive emotion). Thus, negative emotion and ambivalence both reduced defensive responding independent of each other, with the latter accounting for a larger share of the variance in the dependent variables than the former. We return to a discussion of the relative effects of negative emotion and emotional ambivalence in the general discussion section.

Next, to test our second hypothesis, SEM was performed with STATA 14.2, using maximum likelihood parameter estimation, in which defensive responding is modeled as a mediator for the effects of (a) the interaction between implicit bias feedback (coded as "1"; vs. no feedback, coded as "0") and emotional ambivalence on (b) bias awareness. In order to account for clustering of responses within experiments in this mediation analysis, the indirect effect was computed based on the product-of-coefficient approach, using the multilevel mediation analysis command available in STATA that was adapted from Krull and MacKinnon [50]. Subsequently, a bootstrap analysis was performed following the recommendation by Preacher and Hayes [51] with 5000 resampled data sets. Bootstrapping estimates the indirect effect on each resampled data set based on the null hypothesis that the indirect effect is not different from zero. For all analyses below, we reject the null hypothesis if the confidence interval does not include zero [51].

Fig 2 represents the theoretical models and Table 2 summarizes the direct and indirect effects for the hypothesized models. Consistent with our hypothesis, results indicate that the interaction between bias feedback (vs. no feedback) and emotional ambivalence covaried indirectly with bias awareness through its relationship to defensive responding. Conditional indirect effects of implicit bias feedback (vs. no feedback) on awareness were estimated at the 1 SD Above/Below the mean of emotional ambivalence. Results indicate that individuals who experienced high (vs. low) levels of emotional ambivalence responded less defensively to implicit bias feedback, and this, in turn, led to relatively more awareness of bias in the self and others.

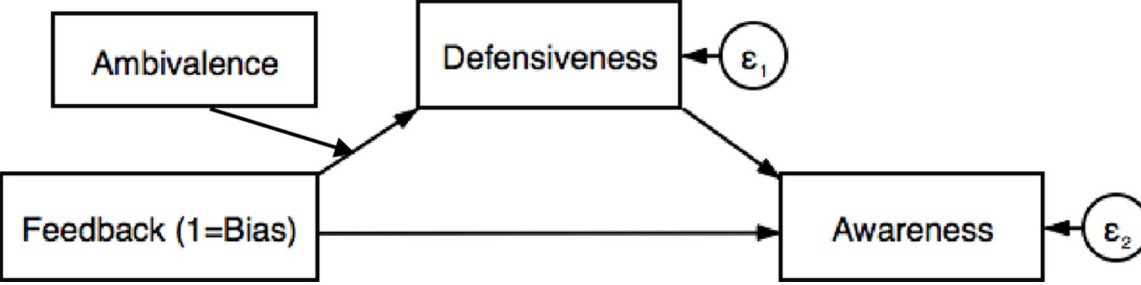

**Fig 2. Conceptual model for SEM analyses.** Conceptual model of the conditional indirect effect of implicit bias feedback (vs. no feedback) on bias awarereness via reduced defensive responding at both high and low levels of emotional ambivalence.

**Table 2. Results from SEM for bias awareness.**

| | b (95% CI) | | | | β | | | | SE | | | | R² |
|---|---|---|---|---|---|---|---|---|---|---|---|---|---|
| **Model** | **FB** | **AMB** | **X** | **DEF** | **FB** | **AMB** | **X** | **DEF** | **SE** | | | | **R²** |
| *Direct* | | | | | | | | | FB | AMB | X | DEF | |
| Bias Aware Defensiveness | .12 (.12, .12)*** .32 (.30, .34)*** | .41 (.41, .41)*** .01 (-.12, .13) | -.23 (-.27, .20)*** -.38 (-.43, -.33)*** | -.32 (-.32, -.32)*** | .33*** .58*** | .43*** .01 *** | -.27*** -.29*** | -.48*** | .01 .002 | .002 .06 | .02 .02 | .001 | .33 .16 |
| *Indirect* | | | | | | | | | | | | | |
| Bias Aware | -.10 (-.11, -.09)*** | -.002 (-.04, .04) | .12 (.10, .14)*** | | -.28*** | -.002 | .14*** | | .00 | .02 | .01 | | |
| *Total* | | | | | | | | | | | | | |
| Bias Aware | .02 (.01, .03)*** | .41 (.37, .45)*** | -.11 (-.14, -.09)** | | .05*** | .43*** | -.13*** | | .01 | .02 | .01 | | |
| *Simple* | FB 1 SD + | | FB 1 SD - | | | | | | FB 1 SD + | | FB 1 SD - | | |
| Bias Aware | -.04 (-.06, -.02)*** | | -.09 (-.12, -.06)*** | | | | | | .01 | | .02 | | |

FB = implicit bias feedback; AMB = emotional ambivalence; X = interaction term between feedback and ambivalence; DEF or Defensiveness = defensive responding; Bias Aware = bias awareness.

FB 1 SD +/- = Effect of feedback 1 SD above/below mean of ambivalence.

SE represent standard error for unstandardized coefficients.

(†p<0.10

*p<0.05

**p<0.01

***p<0.001).

# General discussion

Two studies demonstrate that non-threatening internal conflict–state emotional ambivalence "in the current moment"—attenuates defensiveness to implicit bias feedback. Further, among people high (vs. low) in emotional ambivalence, implicit bias feedback (vs. no feedback) indirectly increases bias awareness through reduced defensive responding. Individuals who experienced high (vs. low) levels of emotional ambivalence following receipt of implicit bias feedback perceived the bias feedback as more valid, credible, and objective, regardless of the extent to which they held implicitly or explicitly prejudicial attitudes. These results may suggest that emotional ambivalence increases bias awareness by creating a mindset in which individual's scope of attention is broadened and their willingness to be influenced by contextual information in increased. As a result, this flexible mindset reduces defensive responding to potentially threatening information, in this case, feedback about the existence of their *own* unconscious racial bias.

The integrative nature of the present research contributes to several areas of psychological inquiry. First, we build on research on attitude ambivalence, internal conflict, and emotional ambivalence, and integrate it with research on defensive responding and prejudice-regulation to demonstrate for the first time that emotional ambivalence is an internal state by which individuals may become more influenced by contextual information, and thus less defensive to potentially self-threatening implicit bias feedback. The current research also helps to advance a growing body of research on the effects of emotional ambivalence on information processing. Considering the evidence that emotional ambivalence can make people more receptive to negative information *about others* when developing perceptions [41], an open and important question was whether it can also make individuals less defensive to negative information *about the self*. Our findings suggest that it does.

Second, that negative affect was also associated with reduced defensive responding following bias feedback is consistent with a large literature demonstrating the role of guilt and negative affect in the self-regulation of prejudiced-responding [10,41,52,53]. Specifically, prior work has shown that *self-directed* negative affect (i.e., guilt) is particularly beneficial for increasing prejudice-regulation among low prejudiced individuals' whereas *other-directed* negative affect (i.e., the desire to avoid general discomfort and social admonishment) decreases prejudicial responding among individuals high in prejudice [41,54,55]. This research has yet to explore whether complex emotions such as emotional ambivalence could motivate reduced defensiveness to bias feedback.

We have shown that negative affect and state experiences of emotional ambivalence both reduce defensive responding independent of each other, with the latter accounting for a larger share of the variance in the dependent variables than the former. Importantly, by studying the state experience of emotional ambivalence we are offering a different route for attenuating defensive responses than has been offered in this prior scholarship. Emotional ambivalence is expected to have such an effect through a different mechanism than guilt. Guilt has been studied in terms of goal pursuit. When people do not reach their egalitarian standards, they feel guilty. Guilt then makes people motivated to reach their standards–to be egalitarian—and one way to do this is to seek goal relevant information and become less defensive to bias feedback and more aware of bias. But, the goal of being egalitarian (induced by guilt) could also be achieved by rejecting the feedback and rejecting evidence of one's bias, hence declaring oneself egalitarian and the feedback wrong. Which of these effects of guilt is found may depend on individual differences. In contrast, we predicted that emotional ambivalence will make people more willing to be influenced by contextual cues and more cognitively flexible. As such, they will become less defensive to influence from expert's knowledge and feedback. Emotional ambivalence does not focus individuals on a particular type of goal or cognition but rather will increase their willingness to be influenced by all available information. By moving beyond the exclusive focus on univalent integral negative affect (e.g., guilt, discomfort) to emotional ambivalence, our research expands the modal conceptualization of internal conflict in the prejudice-regulation literature. As a result, we provide additional nuance about how affect shapes bias awareness.

Further, whereas prior research has established the importance of integral negative emotions, or emotions triggered by the current situation, for prejudice-regulation [41] our research is the first to establish the importance of incidental emotions. Our research is also the first to demonstrate incidental emotional ambivalence as a factor by which high and low prejudiced individuals may become less defensive to threatening bias feedback.

Further, based on prior scholarship, we create dissonance (not ambivalence) in White individuals through delivering bias feedback [17]. This recent scholarship has led to a research interest in how to mitigate the threat of bias feedback so that people can learn from it. This can be done in one of two general and independent ways–by shaping the feedback to be less threatening or by altering the state of the individual to be more receptive to feedback. Some scholarship has focused on that first factor, that is, how to deliver this type of feedback to mitigate the threat [16,17]. Our focus is on the second factor, how internal individual states–emotional ambivalence—can help to prepare people to receive this threatening bias feedback. In doing so, our research advances an emerging literature on defensive responding to implicit bias feedback that has focused on individual differences in accurate awareness of implicit attitudes [8], implicit-explicit incongruence [14], or perception of one's bias relative to others [12]. We are the first to demonstrate that a state experience of emotional internal conflict that is caused by incidental events can increase bias awareness by reducing defensiveness to implicit bias feedback.

## Limitations

The limitations of our current research offer opportunities for future scholarship. First, our theoretical and practical interests concerned how members of high-status social groups (e.g., White people) respond to socially undesirable feedback about their implicit bias towards members of low-status, marginalized social groups (i.e., Black people). As such, we focused our empirical tests on White people only. However, the question of how emotional ambivalence may attenuate defensive responses among non-White people is a necessary direction for future research. Indeed, there are compelling reasons to anticipate that the way in which people respond to feedback indicating bias towards one's own social group (i.e., Black participants learning about implicit bias towards Black people) is a different phenomenon than how people respond to feedback indicating bias towards an out-group (i.e., White participants learning about implicit bias towards Black people).

Additionally, we have suggested that our effects operate through a mechanism that is based on ambivalence amplification theory [22,26] and prior research on internal conflict [23,24]. Response amplification suggests that people who are ambivalent are more influenced by the context than people who are not ambivalent. Research on internal conflicts has shown that creating internal conflict activates a broader general reasoning process that, once accessible, can be applied to subsequent judgments [24,41]. Taken together, this scholarship suggests that the state experience of emotional ambivalence, measured after taking the IAT and receiving feedback (although not manipulated by these activities), should motivate individuals to be more willing to be influenced by others and create a general reasoning process that broadens the scope of information one is willing to consider in new, non-threatening domains. In the current research, we demonstrate the effects of this emotional state on reduced defensiveness. However, future research should extend these observations by measuring the state of emotional ambivalence before these manipulations, or manipulating emotional ambivalence about an unrelated incidental experience [see 41] and measuring cognitive flexibility directly to uncover the psychological mechanism we have theorized.

Future research should also explore the role of emotional ambivalence not only for reducing defensiveness and increasing awareness but also for actual prejudice-regulation and behavior in interracial settings, which we know can be undermined when people respond defensively [12–14,17]. In addition, while the moderating effect of emotional ambivalence on defensive responding was observed in response to feedback that characterized unconscious racial bias in a particularly unflattering manner, future research could examine these effects using less extreme feedback manipulations, such as in interpersonal confrontations (see [10]) or as they operate in relation to other forms of unconscious bias (e.g., gender bias).

These findings point to a simple and promising tool for overcoming defensiveness to self-threatening feedback. The state experience of emotional ambivalence offers an opportunity for reduced defensiveness about prejudice and bias.

## Supporting information

**S1 File.**
(DOCX)

## Author Contributions

**Conceptualization:** Naomi B. Rothman, Joseph A. Vitriol, Gordon B. Moskowitz.

**Data curation:** Joseph A. Vitriol.

**Formal analysis:** Joseph A. Vitriol.

**Investigation:** Naomi B. Rothman, Joseph A. Vitriol.

**Methodology:** Naomi B. Rothman, Joseph A. Vitriol.

**Writing – original draft:** Naomi B. Rothman, Joseph A. Vitriol, Gordon B. Moskowitz.

**Writing – review & editing:** Naomi B. Rothman, Joseph A. Vitriol, Gordon B. Moskowitz.

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
