## [Decision Letter · Decision Letter 0]

28 Aug 2021

PONE-D-21-23604

Internal Conflict and Prejudice-Regulation: Emotional Ambivalence Buffers Against Defensive Responding to Implicit Bias Feedback

PLOS ONE

Dear Dr. Rothman,

Thank you for submitting your manuscript to PLOS ONE. After careful consideration, we feel that it has merit but does not fully meet PLOS ONE’s publication criteria as it currently stands. Therefore, we invite you to submit a revised version of the manuscript that addresses the points raised during the review process.

Both reviewers and I found your manuscript very interesting and of great importance. However, several major concerns were raised. In particular, Reviewer 1 questions your conceptualization of emotional ambivalence and provides an alternative explanation for your findings. Reviewer 2 also requests clarification about the construct of emotional ambivalence. Reviewer 1 also wonders how this work can be reconciled with the dissonance literature, which generally shows that internal conflict does not motivate openness. Reviewer 2 raised important questions about the IAT used in your study. Please provide more explanation of the paradigm used, especially the inclusion of flowers and insects. Were steps taken to ensure that your samples are independent? Finally, please include a limitations section to your discussion.    

We look forward to receiving your revised manuscript.

Kind regards,

Natalie J. Shook

Academic Editor

PLOS ONE

Journal Requirements:

2. Please change "female” or "male" to "woman” or "man" as appropriate, when used as a noun (see for instance https://apastyle.apa.org/style-grammar-guidelines/bias-free-language/gender).

Reviewers' comments:

Reviewer's Responses to Questions

**Comments to the Author**

1. Is the manuscript technically sound, and do the data support the conclusions?

Reviewer #1: No

Reviewer #2: Partly

2. Has the statistical analysis been performed appropriately and rigorously? 

Reviewer #1: No

Reviewer #2: I Don't Know

3. Have the authors made all data underlying the findings in their manuscript fully available?

Reviewer #1: Yes

Reviewer #2: Yes

4. Is the manuscript presented in an intelligible fashion and written in standard English?

Reviewer #1: Yes

Reviewer #2: Yes

5. Review Comments to the Author

Reviewer #1: This is an interesting, important, and timely topic. It’s particularly important to identify ways in which people can receive bias feedback more openly, interrogate their own biases, and, hopefully, behave in more egalitarian ways. But I don’t think this work helps us to identify such ways.

Participants took an IAT and then received (randomly assigned) bias feedback. They then reported their “emotional ambivalence,” defensive reactions to the IAT, and bias awareness. The negative affect items of the ambivalence measure weren’t basic PANAS style items, they were: “angry at myself, guilty, regretful, annoyed at myself, disappointed with myself, shame, self-critical.” That is, they were the very sorts of items that would indicate a non-defensive ownership of one’s bias about which one just received feedback. So it’s no surprise that people who reported greater self-directed negative affect after bias feedback also accepted the IAT results and reported more bias awareness.

Thus, I don’t think their results support the claim that ambivalence makes people more open-minded, less defensive, or more aware of their biases. Instead, it only tells us that the sorts of people who feel bad about their biases also own and admit to them.

Other concerns:

As far as I am aware, there is no evidence that ambivalence or dissonance makes people less defensive (the articles the authors cite that suggest this really aren’t that closely related, e.g., the Rees et al. 2013 paper they cite skirts the issue too, and seems to only consider unimportant judgments like temperature, not ego-threatening ones). Most evidence is to the contrary, and interventions to decrease defensiveness (e.g., self-affirmation) are all about making people feel ok; ambivalence doesn’t feel ok. So a dissonance expert would not agree that “internal conflict … motivates a balanced consideration of multiple different perspectives” (p. 4). Consideration, sure; balanced, probably not. And in the ambivalence literature, “response amplification,” whereby people rather vigorously embrace one of two competing positions, is well documented, so there’s not a lot of evidence of balanced reflection here either. Perhaps the authors are onto conditions under which ambivalence or dissonance reduces defensiveness, but given a vast literature to the contrary, I believe they need to address this issue head-on. What’s different about what they’re doing compared to work where defensiveness increases as a function of inner conflict?

I have implied (without direct evidence) that the negative items on their ambivalence measure are doing the work. The ambivalence formula the authors use has a track record, but it doesn’t allow a test of the possibility that one or the other is driving results. A stronger test would test simple effects of negative and positive affect against the ambivalence score. That is, do the patterns the authors report also obtain when examining only negative affect (independently of positive affect)? If so, we’d be back to concluding that people who feel bad about their biases tend to own and admit to them.

Minor point: the claim is that “people do not recognize prejudice in…themselves,” (p. 3), but several studies show that people are in fact aware of their implicitly measured prejudices. They may be unaware of the impact of those prejudices on judgments/behavior, but they are aware of the prejudices themselves.

Reviewer #2: This manuscript examines the relationship between emotional ambivalence and defensive responding to negative feedback in the context of implicit racial bias. The predicted relationship between ambivalence and defensiveness is both intuitive and theoretically-grounded, and the manuscript is well-written. That said, the manuscript can be improved upon in several ways, which I elaborate upon below largely in the order in which they appear in the manuscript.

I’m fairly well-versed in the attitudinal ambivalence literature, but I know less about emotional ambivalence. It would be helpful for the authors to more clearly articulate how they conceptualize emotional ambivalence. Do they understand it to be a trait or a state? I bring up this point because they measured emotional ambivalence after participants completed the IAT and received (or did not receive) feedback. Do the authors understand participants’ emotional ambivalence to reflect that feedback, or is emotional ambivalence a stable disposition of participants? Given that the authors report that there was no main effect of feedback on emotional ambivalence, I assume the latter, but it would be helpful for this point to be clarified. This point would also seem to have implications for application: Do we need to seek out highly emotionally ambivalent people as good candidates for bias reduction, or can we induce people to be emotionally ambivalent to make them more susceptible for bias reduction?

The authors note that they removed non-White participants from their samples “as planned”. I would like for them to elaborate on why they planned to remove non-White participants, and also report whether the results changed when all participants, or at least non-White non-Black participants, are included in the sample. (I carve out this caveat for Black participants given the content of the feedback. It seems reasonable to that Black people would respond differently, e.g., skeptically, to feedback that they are biased against Black people).

OK here’s a really important point: I need for the authors to more clearly describe the IAT that they used, and to articulate the logic behind their choice of stimuli. I would characterize myself as highly familiar with the implicit social cognition literature, and with the IAT specifically. I’ve never heard of an IAT with Black people, White people, flowers, insects, positive attributes, and negative attributes as stimuli. In fact, I can’t even wrap my head around what it might mean to have “pro-White/anti-Black *or* pro-Flower/anti-Insect bias. Without sufficient rationale, this would seem to be a serious threat to construct validity. That said, I understand that the authors intended to use this IAT deceptively, but there are still problems with this approach. They state that “…our goal is to have participants believe it (the feedback) is accurate” (p. 8), but do they have any evidence that participants believed the IAT was legitimate? I mean, I guess that people believed the bogus pipeline, so maybe they believe a Black White Flower Insect IAT measures racial bias? Still, some evidence that the paradigm worked as intended would be helpful. Additionally, the authors used IAT scores as control variables. However, given my serious concerns about the content validity of this IAT, it’s unclear to me what exactly they are controlling for. At minimum, I’m curious to see whether the pattern of results replicate without controlling for scores on this strange IAT.

I’ve patted myself on the back twice in this review so far, indicating literatures with which I am fairly and highly familiar. Now I have to disclose that I am not familiar enough with either of the statistical methods the authors rely on in this manuscript (i.e., MLM, SEM) to critically evaluate their use here. So I won’t comment on that section of the manuscript…

…with one exception. On p.12, the statistics the authors report in text for the analyses 1 SD above and below the mean on emotional ambivalence would seem to be backwards: the Z value at 1 SD below the mean is smaller than the Z value at 1 SD above the mean, but the text describes the opposite pattern of results. Am I confused?

And finally, while I can clearly understand the authors’ rationale for why emotional ambivalence would reduce defensiveness, its less clear to me why emotional ambivalence would also be related to bias awareness. The authors need to make a more compelling case for why we should expect this outcome, and why bias awareness matters in the big picture.

Minor points:

On p.7, the authors state that “…we were able to obtain a large, non-random sample of White Americans with sufficient variability on demographic characteristics…” It’s unclear to me what ‘sufficient’ refers to in this case. Sufficient for what?

Also on p.7, they refer to ‘non-Whites’. Ostensibly there are people in there, so I would encourage the authors to use racial labels as adjectives rather than nouns, e.g., “non-White participants”.

6. PLOS authors have the option to publish the peer review history of their article (what does this mean?). If published, this will include your full peer review and any attached files.

Reviewer #1: No

Reviewer #2: No

---

## [Author Response · Author response to Decision Letter 0]

3 Nov 2021

Dear Dr. Shook, 

Thank you for inviting us to revise and resubmit the manuscript (PONE-D-21-23604) entitled Internal Conflict and Prejudice-Regulation: Emotional Ambivalence Buffers Against Defensive Responding to Implicit Bias Feedback for publication consideration in PLOS ONE. We have addressed each comment in the revised manuscript and have responded to each point raised by yourself and the reviewer(s) below. 

Comments from Editor

Thank you for submitting your manuscript to PLOS ONE. After careful consideration, we feel that it has merit but does not fully meet PLOS ONE’s publication criteria as it currently stands. Therefore, we invite you to submit a revised version of the manuscript that addresses the points raised during the review process.

Our Response: Thank you for this opportunity to revise our manuscript. 

Both reviewers and I found your manuscript very interesting and of great importance. However, several major concerns were raised. 

In particular, Reviewer 1 questions your conceptualization of emotional ambivalence and provides an alternative explanation for your findings. 

Reviewer 2 also requests clarification about the construct of emotional ambivalence. 

Our Response: We have clarified that high ambivalence does not indicate “people who feel bad about their biases” as suggested by Reviewer 1, but rather, “people who feel emotionally ambivalent in the present moment”. We conceptualized emotional ambivalence as a state experience of positive and negative emotions “in the present moment” and triggered by prior, unrelated experiences. We build on the work on Rees et al., (2013) who study the effects of incidental emotional ambivalence. The emotions literature distinguishes between integral emotions, emotions triggered by the current situation, and incidental emotions, emotions triggered by a prior, unrelated experience (Lerner & Keltner, 2001). Integral emotions are likely to exert a stronger influence on openness to influence than incidental emotion because integral emotions are generated from the decision context itself and are more likely than incidental emotions to be infused into the decision process. Our main focus on incidental emotions serves as a more conservative test of the role of emotional ambivalence on openness to influence. Consistent with our focus on incidental emotions, we did not predict (and did not find) that emotional ambivalence would be caused by the feedback itself.

Reviewer 1 also wonders how this work can be reconciled with the dissonance literature, which generally shows that internal conflict does not motivate openness. 

Our Response: We thank Reviewer 1 for raising this concern and challenging us to think more carefully about our conceptualization of emotional ambivalence and its relationship to other perspectives on internal conflict. We provide a comprehensive response to this point below, in which we clarify the critical differences in our approach that we believe create the conditions under which ambivalence attenuates defensive responses to bias feedback (rather than increase defensive responses to bias feedback). We have also included a discussion of this important point in our revised manuscript, in the section now titled, “Differences between Attitude Ambivlaence and Emotional Ambivalence”

Reviewer 2 raised important questions about the IAT used in your study. Please provide more explanation of the paradigm used, especially the inclusion of flowers and insects. 

Our Response: We apologize. This was an error on our part. The description we provided (e.g., flowers and insects) included information from the write-up of a different paper results/methods second. We have cut this out, and made sure there are no additional errors in the manuscript. 

Were steps taken to ensure that your samples are independent? 

Our Response: As in all experiments, the samples are independent and participants were recruited to be in one group of the experiment are prevented from participating in another. To be reimbursed each person needs to have a unique identification code, and one is prevented from signing up for the experiment a second time using the code as a wall. Of course, someone could always establish multiple identities, but this is true of any experiment where a person could produce a false ID if so motivated. 

Finally, please include a limitations section to your discussion. 

Our Response: We apologize for the omission and have added in a limitations section where we have identified limitations and opportunities for future scholarship. 

Reviewer #1 comments 

This is an interesting, important, and timely topic. It’s particularly important to identify ways in which people can receive bias feedback more openly, interrogate their own biases, and, hopefully, behave in more egalitarian ways. But I don’t think this work helps us to identify such ways.

Participants took an IAT and then received (randomly assigned) bias feedback. They then reported their “emotional ambivalence,” defensive reactions to the IAT, and bias awareness. The negative affect items of the ambivalence measure weren’t basic PANAS style items, they were: “angry at myself, guilty, regretful, annoyed at myself, disappointed with myself, shame, self-critical.” That is, they were the very sorts of items that would indicate a non-defensive ownership of one’s bias about which one just received feedback. So it’s no surprise that people who reported greater self-directed negative affect after bias feedback also accepted the IAT results and reported more bias awareness. Thus, I don’t think their results support the claim that ambivalence makes people more open-minded, less defensive, or more aware of their biases. Instead, it only tells us that the sorts of people who feel bad about their biases also own and admit to them.

Our Response: There are a few separate points embedded in this comment, and we will respond to each, in turn. However, the most important thing to state first is that our prediction and finding is not that people who feel more negative affect will accept the IAT feedback and have more bias awareness. The reviewer would be correct in stating that all we would be showing is that the sort of people who feel bad are the sort of people who own their bias, if all that we predicted and found was an increase in negative emotions led to increases in bias awareness. Such an observation is consistent with a large literature demonstrating the role of guilt and negative affect in the self-regulation of prejudiced-responding (e.g., Devine, Forscher, Austin, & Cox, 2012; Czopp Monteith & Mark, 2006; Monteith, Ashburn-Nardo, Voils, & Czopp, 2002; Moskowitz, Gollwitzer, Wasel, & Schaal, 1999). 

But this is not what we predict. It is not just increases in negative emotions that have the effect of reducing defensive responses. Rather, while controlling for both positive and negative affect, we demonstrate that ambivalence is uniquely related to reduced defensive responding and accounts for a larger share of variance in defensive responding than univalent affect. On p.17-18, we now dedicate substantial space to addressing the role of negative affect in our analysis and return to this discussion in our general discussion section. We thank you for encouraging us to clarify the important differences between our work and the prior research on negative emotions.

Next, we turn to the additional points raised by the reviewer. First, you are accurate that in these studies, we have measured emotional ambivalence after participants received (or did not receive, depending on condition) bias feedback. As we describe, we use established methods for measuring and calculating objective emotional ambivalence. Specifically, we had participants report their positive emotions and negative emotions, separately. We used a positive affect measure with items that include both higher activation/arousal positive emotions (e.g., excited, energetic) and lower activation positive emotions (e.g., content, good), rather than just PANAS items that are all high activation. Our measure of negative affect is based on the work by Monteith and colleagues (e.g., Monteith, Devine, & Zuwerink, 1993; Monteith et al., 2002; Monteith, & Mark 2005), some of the only work that examines emotional responses and their relationship with bias awareness and prejudice-regulation. 

Then, as is standard in the literature, we utilized the Griffins formula to calculate both the similarity and extremity in the coexisting positive and negative components. The important point is that the calculation takes into consideration the strength/extremity of both positive and negative emotional reactions, simultaneously, as well as the similarity/difference in positive and negative substrates. People can be anywhere on the continuum from low ambivalence (e.g., high negative-low positive; low negative-high positive) to high ambivalence (e.g., high negative-high positive), and we are particularly interested in the benefits of people feeling high levels of emotional ambivalence for their subsequent cognitive processing (e.g., defensiveness). The point is that high ambivalence indicates coexistence of both positive and negative affective states.

To clarify, high ambivalence does not indicate “people who feel bad about their biases” but rather, “people who feel emotionally ambivalent in the present moment” about some prior, unrelated event. We conceptualized emotional ambivalence as a state experience of positive and negative emotions “in the present moment” and, importantly, triggered by prior, unrelated experiences. We build on the work on Rees et al., (2013) who study the effects of incidental emotional ambivalence. The emotions literature distinguishes between integral emotions, emotions triggered by the current situation, and incidental emotions, emotions triggered by a prior, unrelated experience (Lerner & Keltner, 2001). Integral emotions are likely to exert a stronger influence on openness to influence than incidental emotion because integral emotions are generated from the decision context itself and are more likely than incidental emotions to be infused into the decision process. Our main focus on incidental emotions serves as a more conservative test of the role of emotions on openness to influence. Consistent with our focus on incidental emotions, we did not predict (and did not find) that emotional ambivalence would be caused by the feedback itself. We have clarified this point significantly in the revised manuscript in response to your questions and feedback. 

Second, because we are conceptualizing this as incidental emotional ambivalence, we did not predict that bias feedback would induce emotional ambivalence. And, we found no evidence that bias feedback did induce emotional ambivalence. RACE IAT D-Scores are not correlated with emotional ambivalence in (.04; See Table 1). So, we don’t think it’s accurate to describe the phenomenon as “feeling bad about their biases”. For two independent reasons. The phenomenon is not about feeling bad but feeling ambivalent. And the feelings are not about the feedback.

Third, we predict that bias feedback will increase defensiveness, and emotional ambivalence will moderate or attenuate this defensive response. Indeed, this is what we find. Thus, it is reasonable to conclude that people who experienced high (vs. low) levels of emotional ambivalence in that moment (which means they feel both high levels of positive and negative emotions at the same time) are less defensive to bias feedback compared to people who experienced low levels of emotional ambivalence (e.g., they are not feeling both at the same time to the same extent). 

Other Reviewer 1 Concerns:

As far as I am aware, there is no evidence that ambivalence or dissonance makes people less defensive (the articles the authors cite that suggest this really aren’t that closely related, e.g., the Rees et al. 2013 paper they cite skirts the issue too, and seems to only consider unimportant judgments like temperature, not ego-threatening ones). Most evidence is to the contrary, and interventions to decrease defensiveness (e.g., self-affirmation) are all about making people feel ok; ambivalence doesn’t feel ok. So a dissonance expert would not agree that “internal conflict … motivates a balanced consideration of multiple different perspectives” (p. 4). Consideration, sure; balanced, probably not. And in the ambivalence literature, “response amplification,” whereby people rather vigorously embrace one of two competing positions, is well documented, so there’s not a lot of evidence of balanced reflection here either. Perhaps the authors are onto conditions under which ambivalence or dissonance reduces defensiveness, but given a vast literature to the contrary, I believe they need to address this issue head-on. What’s different about what they’re doing compared to work where defensiveness increases as a function of inner conflict?

Our Response: First, we agree there is no evidence that ambivalence makes people less defensive. We also agree that the Rees et al., (2013) paper focuses on non-threatening decisions, so it’s a different effect than what we are showing here. And further, we agree that most interventions (e.g., self-affirmation) are all about making people feel okay and ambivalence probably does not make people feel okay. This gap in the literature is precisely the one we are addressing.

We appreciate your inquiry about what is different about what we are doing compared to other work that has suggested the opposite. Below (and in the revised manuscript) we clarify the critical differences in our approach that we believe create the conditions under which ambivalence attenuates defensive responses to bias feedback (rather than increase defensive responses to bias feedback).

First, based on prior scholarship, we create dissonance in White individuals through delivering bias feedback (e.g., Vitriol & Moskowitz, 2021). This recent scholarship has led to a research interest in how to mitigate the threat of bias feedback so that people can learn from it. This can be done in one of two general and independent ways – shaping the feedback to be less threatening, altering the state of the individual to be more receptive to feedback. Some scholarship has focused on that first factor, how to deliver this type of feedback to mitigate the threat (Vitriol & Moskowitz, 2021; Moskowitz & Vitriol, 2022). Our focus is on the second factor, how internal individual states – emotional ambivalence - can help to prepare people to receive this threatening bias feedback. We believe emotional ambivalence can do this because it increases individuals’ openness to influence by contextual information (e.g., Katz & Glass, 1979).

Second, traditional approaches to the study of attitude ambivalence build on the dissonance literature, and thus largely assume that ambivalence represents a threatening consistency violation, can be experienced as unpleasant, and lead to negative affect. However, ambivalence does not always produce negative affect or threat. Ambivalence, for instance, can be desirable when an issue is controversial (Maio and Haddock, 2004) can be negatively related to physiological arousal (Maio, Greenland, Bernard & Esses, 2001) and is even cultivated by people in the face of uncertainty in regard to obtaining a desirable outcome (Reich & Wheeler, 2016). As such, ambivalence does not always need to be threatening. 

Dissonance is usually the result of behavioral commitment that is in conflict with a pre-existing attitude. The conflict with an attitude, often one that is important to the self, creates threat. Steele calls this a threat to the integrity of the self-system. Thus, dissonance research is about a threat to the integrity of the self-system and is always about removing that threat, with defensiveness being one prominent strategy. In our studies of emotional ambivalence, people do not have to make an unequivocal stance (there is no behavioral commitment), and so we do not believe that emotional ambivalence is experienced as dissonance. There is no threat, and the integrity of the self-system is never challenged. While ambivalence is defined by conflict, it occurs either within one’s attitude but is not related to one’s behavioral commitment (attitude ambivalence) or within one’s emotions and also not related to one’s behavioral commitment (emotional ambivalence). In this way, the emotional ambivalence in our studies remain noncommittal, and so ambivalence is not expected to become particularly unpleasant or threatening (for the reverse argument see Van Harreveld, van der Pligt et al., 2009). 

Certainly, the feedback we give people is dissonance arousing. It is intended to. Our concern is with what mitigates feeling defensive in response to that threat. But the emotional ambivalence people experience is not necessarily dissonance arousing. This is why the reviewer’s concern about dissonance always increasing defensiveness is not troublesome to us. We are not studying dissonance as a mitigation tool to dissonance. We are studying emotional ambivalence, a non-threatening state of harboring mixed emotions that promotes openness to influence, not defense of an existing belief, as a means of reducing the dissonance aroused by the feedback. The existing state of having both positive and negative emotions was hypothesized to motivate resolution, but it does so by calling for openness to both sides, not defending one previously existing (and strongly committed) position, as is commonly observed in dissonance-reduction paradigms.

Third, we agree that ambivalence amplification theory is relevant. But we believe our findings are consistent with response amplification findings. This theory suggests that people who are ambivalent are more influenced by the context than people who are not ambivalent (e.g., Katz & Glass, 1979; Katz & Haas, 1988; MacDonald & Zanna, 1998). For instance, research has demonstrated that ambivalent people are more likely to react positively or negatively toward a member of a stigmatized group depending on whether positive or negative situational cues were first provided about a member of that social group, respectively. Specifically, people who have ambivalent attitudes are more likely to exhibit extreme positive or negative behavioral intentions than their unambivalent counterparts, depending on the type of information that they attend to. In contrast, these situational cues do not impact unambivalent participants (MacDonald & Zanna, 1998). By extension, we believe that emotional ambivalence will make people more open to influence by contextual information provided by the experimenter about their race bias. 

It’s important to note that our findings are not inconsistent with research that has shown that individuals who are high in attitude ambivalence are more open to persuasion. That is, for example, they can be persuaded by individuals from both their own university and another university, whereas low ambivalent individuals are less open to persuasion, only demonstrating attitude change when the source of the message is the student at their own university (Zemborian & Johar, 2007). Our work advances this scholarship by demonstrating that emotional ambivalence can make people more open to influence from others, even when that information and knowledge provided is potentially self-threatening. 

I have implied (without direct evidence) that the negative items on their ambivalence measure are doing the work. The ambivalence formula the authors use has a track record, but it doesn’t allow a test of the possibility that one or the other is driving results. A stronger test would test simple effects of negative and positive affect against the ambivalence score. That is, do the patterns the authors report also obtain when examining only negative affect (independently of positive affect)? If so, we’d be back to concluding that people who feel bad about their biases tend to own and admit to them.

Our Response: The results we report concerning the interaction between ambivalence and bias feedback control for the univalent negative and positive affect in the model. We agree that we can more convincingly demonstrate the unique role of ambivalent affect, so we now note on p.17-18:

“We also conducted an analysis in which we estimated the simultaneous effect of three interaction terms --- feedback condition and positive emotion, negative emotion, or ambivalent emotion -- on defensive responding, again using a multilevel model with maximum likelihood estimation and sample submitted as a random-intercept term. Doing so renders all interaction terms statistically non-significant (ps > .05), which is to be expected, given that ambivalence is computed using positive and negative emotion. Nonetheless, we estimated two models in which we regressed defensive responding on positive, negative, and ambivalent emotion, separately for participants in the Bias Feedback and No Feedback condition. Again, we used multilevel models with maximum likelihood estimation and sample submitted as a random-intercept term. For participants in the No Feedback condition, positive emotion significantly predicted defensive responding (b =-.27, SE = .07, (95% CI = -.41, -.13), p < .001), but neither negative emotion (b = -.05, SE = .12, (95% CI = -.29, .19), p = .69) nor ambivalent emotion (b = -.08, SE = .14, (95% CI = -.35, .19), p = .54) was a significant predictor. In contrast, for participants in the Bias Feedback condition, positive emotion did not significantly predict defensive responding (b = -.10, SE = .06, (95% CI = -.22, .01), p = .08). However, both negative emotion (b =-.20, SE = .08, (95% CI = -.35, -.04), p = .013) and ambivalent emotion (b = -.24, SE = .10, (95% CI = -.44, -.04), p = .02) were significant predictors. 

More importantly, because all variables were rescaled to run 0-1, we are able to compare the strength of the relationship between defensive responding and negative or ambivalent emotion in the Bias Feedback condition. Inspection of the unstandardized coefficients indicates that moving from the lowest to the highest levels of negative emotion corresponded with approximately 20% reduction in defensive responding (while controlling for ambivalent emotion and positive emotion), whereas moving from the lowest to the highest levels of ambivalence corresponded with approximately 24% reduction in defensive responding (while controlling for negative and positive emotion). Thus, negative emotion and ambivalence both reduced defensive responding independent of each other, with the latter accounting for a larger share of the variance in the dependent variables than the former. We return to a discussion of the relative effects of negative emotion and emotional ambivalence in the general discussion section.

We have added the following to our discussion section (pp. 22-23)

Second, that negative affect was also associated with reduced defensive responding following bias feedback is consistent with a large literature demonstrating the role of guilt and negative affect in the self-regulation of prejudiced-responding (e.g., Devine, Forscher, Austin, & Cox, 2012; Czopp Monteith & Mark, 2006; Monteith, Ashburn-Nardo, Voils, & Czopp, 2002; Moskowitz, Gollwitzer, Wasel, & Schaal, 1999). Specifically, prior work has shown that self-directed negative affect (i.e., guilt) is particularly beneficial for increasing prejudice-regulation among low prejudiced individuals’ whereas other-directed negative affect (i.e., the desire to avoid general discomfort and social admonishment) decreases prejudicial responding among individuals high in prejudice (e.g., Monteith, Devine, & Zuwerink, 1993; Monteith et al., 2002; Monteith, & Mark 2005). This research has yet to explore whether complex emotions such as emotional ambivalence could motivate increased openness and receptivity to bias feedback. We have shown that negative affect and incidental emotional ambivalence both reduce defensive responding independent of each other, with the latter accounting for a larger share of the variance in the dependent variables than the former. Importantly, by studying incidental emotional ambivalence we are offering a different route for attenuating defensive responses than has been offered in this prior scholarship. Emotional ambivalence is expected to have such an effect through a different mechanism than guilt. Guilt has been studied in terms of goal pursuit. When people do not reach their egalitarian standards, they feel guilty. Guilt then makes people motivated to reach their standards – to be egalitarian - and one way to do this is to seek goal relevant information and become less defensive to bias feedback and more open to awareness of bias. But, the goal of being egalitarian (induced by guilt) could also be achieved by rejecting the feedback and rejecting evidence of one’s bias, hence declaring oneself egalitarian and the feedback wrong. Which of these effects of guilt is found may depend on individual differences. In contrast, we predicted that emotional ambivalence will make people more open to influence by contextual cues (Katz & Glass, 1979) and being more cognitively flexible. As such, they will become more receptive to influence from expert’s knowledge and feedback. Emotional ambivalence does not focus individuals on a particular type of goal or cognition but rather will open them up to influence by all available information. By moving beyond the exclusive focus on univalent integral negative affect (e.g., guilt, discomfort) to emotional ambivalence, our research expands the modal conceptualization of internal conflict in the prejudice-regulation literature. By doing so, we provide additional nuance about how affect shapes bias awareness. 

Minor point: the claim is that “people do not recognize prejudice in…themselves,” (p. 3), but several studies show that people are in fact aware of their implicitly measured prejudices. They may be unaware of the impact of those prejudices on judgments/behavior, but they are aware of the prejudices themselves.

Our Response: We actually wrote “people often do not recognize” not “people do not”. We have added the word “most”. And for those who do, most do not know how it influences them. While some people may show some level of awareness about their scores on implicit measures (e.g., Hahn), many do not. Further, much research in this domain indicates that implicit and explicit attitudes have independent effects on judgment and behavior in socially sensitive domains (e.g., Kurdi et al., 2019). That defensive reactions to implicit bias feedback commonly elicit defensive responses (e.g., Howell et al., 2017; Vitriol & Moskowitz, 2021) and research also indicates that the results of the IAT are often in conflict with beliefs about one’s explicit attitudes, egalitarian norms, and self-concepts. Together, these findings demonstrate that many people are unaware of both their implicit attitudes and its influence on behavior.

Reviewer #2 comments

 This manuscript examines the relationship between emotional ambivalence and defensive responding to negative feedback in the context of implicit racial bias. The predicted relationship between ambivalence and defensiveness is both intuitive and theoretically-grounded, and the manuscript is well-written. That said, the manuscript can be improved upon in several ways, which I elaborate upon below largely in the order in which they appear in the manuscript.

Our Response: Thank you for suggestions in this review. 

I’m fairly well-versed in the attitudinal ambivalence literature, but I know less about emotional ambivalence. It would be helpful for the authors to more clearly articulate how they conceptualize emotional ambivalence. Do they understand it to be a trait or a state? I bring up this point because they measured emotional ambivalence after participants completed the IAT and received (or did not receive) feedback. Do the authors understand participants’ emotional ambivalence to reflect that feedback, or is emotional ambivalence a stable disposition of participants? Given that the authors report that there was no main effect of feedback on emotional ambivalence, I assume the latter, but it would be helpful for this point to be clarified. This point would also seem to have implications for application: Do we need to seek out highly emotionally ambivalent people as good candidates for bias reduction, or can we induce people to be emotionally ambivalent to make them more susceptible for bias reduction?

Our Response: We conceptualized emotional ambivalence as a state experience of positive and negative emotions “in the present moment” and triggered by prior, unrelated experiences (see Rees et al., 2013). The emotions literature distinguishes between integral emotions, emotions triggered by the current situation, and incidental emotions, emotions triggered by a prior, unrelated experience (Lerner & Keltner, 2001). Integral emotions are likely to exert a stronger influence on openness to influence than incidental emotion because integral emotions are generated from the decision context itself and are more likely than incidental emotions to be infused into the decision process. Our main focus on incidental emotions serves as a more conservative test of the role of emotions on openness to influence. Consistent with our focus on incidental emotions, we did not predict (and did not find) that emotional ambivalence would be caused by the feedback itself. 

The authors note that they removed non-White participants from their samples “as planned”. I would like for them to elaborate on why they planned to remove non-White participants, and also report whether the results changed when all participants, or at least non-White non-Black participants, are included in the sample. (I carve out this caveat for Black participants given the content of the feedback. It seems reasonable to that Black people would respond differently, e.g., skeptically, to feedback that they are biased against Black people).

Our Response: We agree that examining the experience of defensiveness, and the effectiveness of strategies like emotional ambivalence at mitigating defensiveness, among non-White respondents is an interesting and important research question. However, we are unable to examine that research question using the current data, as we have too few non-White respondents in our samples, rendering tests involving these participants statistically under-powered and results of these tests non-diagnostic of our hypotheses. Further, our theoretical and practical interests concern how members of high-status social groups (e.g., White people) respond to socially undesirable feedback about their implicit bias towards members of low-status, marginalized social groups (i.e., Black people). Such status differences between these social groups likely contribute to critical theoretical and empirical differences regarding the psychological dynamics that underpin (a) responses to feedback about one’s implicit attitudes, (b) perceptions and interpretations of prejudice and discrimination in society, (c) and the internalization of egalitarian norms and beliefs about of one’s personal responsibility in regulating bias. Finally, there are compelling reasons to anticipate that the way in which people respond to feedback indicating bias towards one’s own social group (i.e., Black participants learning about implicit bias towards Black people) is a different phenomenon than how people respond to feedback indicating bias towards an out-group (i.e., White participants learning about implicit bias towards Black people). 

In short, examining our research questions among non-White participants is not so simple as testing the same hypotheses on a subset of our participants or on the entire sample; (a) we lack sufficient statistical power to examine non-White respondents separately, (b) are theoretically and practically concerned with White respondents reactions to feedback about bias towards Black people, and (c) believe that this phenomena is quite different for perceiver and targets, low and high-status group members, and in regards to feedback about implicit bias towards the in-group compared to the out-group.

However, because we agree that examining our research questions among non-White participants is an important research question, and because our inability to do that here is a major limitation, we discuss this as a future direction that future research should undertake on p.23.

OK here’s a really important point: I need for the authors to more clearly describe the IAT that they used, and to articulate the logic behind their choice of stimuli. I would characterize myself as highly familiar with the implicit social cognition literature, and with the IAT specifically. I’ve never heard of an IAT with Black people, White people, flowers, insects, positive attributes, and negative attributes as stimuli. In fact, I can’t even wrap my head around what it might mean to have “pro-White/anti-Black *or* pro-Flower/anti-Insect bias. Without sufficient rationale, this would seem to be a serious threat to construct validity. That said, I understand that the authors intended to use this IAT deceptively, but there are still problems with this approach. They state that “…our goal is to have participants believe it (the feedback) is accurate” (p. 8), but do they have any evidence that participants believed the IAT was legitimate? I mean, I guess that people believed the bogus pipeline, so maybe they believe a Black White Flower Insect IAT measures racial bias? Still, some evidence that the paradigm worked as intended would be helpful. Additionally, the authors used IAT scores as control variables. However, given my serious concerns about the content validity of this IAT, it’s unclear to me what exactly they are controlling for. At minimum, I’m curious to see whether the pattern of results replicate without controlling for scores on this strange IAT.

Our Response: This was an error on our part! The reviewer does know the literature and there is no strange flower, insect, Black, White IAT. We used the standard Black-White IAT and mistakenly included descriptions of a totally separate flower-insect IAT used in a different experiment. We are embarrassed to have allowed that error to go undetected in our methods section and have now fixed it.

I’ve patted myself on the back twice in this review so far, indicating literatures with which I am fairly and highly familiar. Now I have to disclose that I am not familiar enough with either of the statistical methods the authors rely on in this manuscript (i.e., MLM, SEM) to critically evaluate their use here. So I won’t comment on that section of the manuscript…

…with one exception. On p.12, the statistics the authors report in text for the analyses 1 SD above and below the mean on emotional ambivalence would seem to be backwards: the Z value at 1 SD below the mean is smaller than the Z value at 1 SD above the mean, but the text describes the opposite pattern of results. Am I confused?

Our Response: Z statistics (like t) is not an indication of effect size, but is a function of standard error. The analysis conducted at 1 SD below the mean has a larger standard error than the analysis at 1 SD above the mean, hence differences in Z that are counter-intuitive. To understand the effect size of the bias feedback condition at 1 SD above or below the mean, one should inspect the unstandardized coefficients, given that all variables were rescaled to run from 0-1. As we note on p.16:

“substantively these estimates indicate that for participants with low levels of emotional ambivalence, bias feedback (vs. no feedback) led to approximately a 39% increase in defensive responding; for participants with relatively high levels of emotional ambivalence, bias feedback (vs. no feedback) led to approximately a 25% increase in defensive responding.

Inspection of the standardized coefficients lead to the same conclusion. Also, we graph these effects in Figure 1 to strengthen readers’ intuitions about our results.

And finally, while I can clearly understand the authors’ rationale for why emotional ambivalence would reduce defensiveness, its less clear to me why emotional ambivalence would also be related to bias awareness. The authors need to make a more compelling case for why we should expect this outcome, and why bias awareness matters in the big picture.

Our Response: Bias awareness reflects a person’s understanding that they have bias. One would not be motivated to change one’s behavior without accepting culpability. Our argument is that defensiveness prevents people from accepting that they have bias. The feedback tells them they do, the defensive response renders the feedback wrong or illegitimate, and hence blocks one from seeing bias in the self. Without bias awareness, there is no impetus to change. Thus, the role of emotional ambivalence is to remove the obstacle to bias awareness, defensiveness. With that obstacle removed, the feedback can be received and accepted. This gives it the power to motivate self-regulation. We have included the following improved theorizing in the revised manuscript (pp. 9)

Furthermore, we propose that reduced defensiveness will increase concern about and awareness of one’s implicit racial bias. Because people who are high in bias awareness are attuned to the possibility that they exhibit subtle biases (Perry et al., 2015), this awareness should be higher among individuals who are less (vs. more) defensiveness. Thus, we expected a conditional indirect effect of implicit bias feedback (vs. no feedback) on bias awareness through reduced defensive responding, with individuals high (vs. low) in emotional ambivalence showing reduced defensive responding and, in turn, increased bias awareness. Examining reactions to implicit racial bias feedback is a fruitful domain for testing whether emotional ambivalence increases receptivity to threatening information about the self, and thus greater awareness of bias. To our knowledge, emotional ambivalence has not been examined in the prejudice-regulation domain. 

Minor points:

On p.7, the authors state that “…we were able to obtain a large, non-random sample of White Americans with sufficient variability on demographic characteristics…” It’s unclear to me what ‘sufficient’ refers to in this case. Sufficient for what?

Our Response: As Paolacci and Chandler (2014) note, which we cite in the main text, “In general, workers are diverse but not representative of the populations they are drawn from… In sum, the pool of available workers is large and diverse. It can replace or supplement traditional convenience samples, but it should not be treated as representative of the general population. The sheer size of the MTurk workforce and the possibility to selectively recruit workers…can also make it useful to reach samples with specific characteristics”

Also on p.7, they refer to ‘non-Whites’. Ostensibly there are people in there, so I would encourage the authors to use racial labels as adjectives rather than nouns, e.g., “non-White participants”.

Our Response: This is an excellent point and we have sought to revise our language to reflect this perspective throughout the manuscript.

---

## [Decision Letter · Decision Letter 1]

19 Dec 2021

PONE-D-21-23604R1Internal Conflict and Prejudice-Regulation: Emotional Ambivalence Buffers Against Defensive Responding to Implicit Bias FeedbackPLOS ONE

Dear Dr. Rothman,

Thank you for submitting your manuscript to PLOS ONE. After careful consideration, we feel that it has merit but does not fully meet PLOS ONE’s publication criteria as it currently stands. Therefore, we invite you to submit a revised version of the manuscript that addresses the points raised during the review process. I greatly appreciate your responsiveness to the first round of reviews. You have done a very thorough job of addressing the reviewers' and my initial concerns, and the manuscript is much stronger. However, a point of clarification has come up with the revised manuscript, regarding the proposition that emotional ambivalence in the current studies is incidental. As the assessment of positive and negative emotion occurred right after receiving bias feedback (or not) and the IAT, it is hard to believe that participants' current emotions were unrelated to study procedures. Ideally, incidental emotions would have been assessed prior to the manipulation. Although the emotional ambivalence index did not differ by condition, did positive or negative emotion differ by condition? A stronger case for emotions being incidental in these studies needs to be made, and the concern about the procedure should be noted in the limitations.      

We look forward to receiving your revised manuscript.

Kind regards,

Natalie J. Shook

Academic Editor

PLOS ONE

Reviewers' comments:

Reviewer's Responses to Questions

**Comments to the Author**

1. If the authors have adequately addressed your comments raised in a previous round of review and you feel that this manuscript is now acceptable for publication, you may indicate that here to bypass the “Comments to the Author” section, enter your conflict of interest statement in the “Confidential to Editor” section, and submit your "Accept" recommendation.

Reviewer #1: (No Response)

2. Is the manuscript technically sound, and do the data support the conclusions?

Reviewer #1: Partly

3. Has the statistical analysis been performed appropriately and rigorously? 

Reviewer #1: I Don't Know

4. Have the authors made all data underlying the findings in their manuscript fully available?

Reviewer #1: Yes

5. Is the manuscript presented in an intelligible fashion and written in standard English?

Reviewer #1: No

6. Review Comments to the Author

Reviewer #1: I was perhaps a bit too dismissive of the ambivalence angle and instead fixating on mere negative affect in my first round review. I appreciate that the authors took my confusion as an opportunity to clarify their logic, especially regarding incidental vs. integral affect. However, now I see a clearer disconnect between the logic laid out in the intro and the methods. There’s nothing “incidental” about the emotions people experience immediately after IAT bias feedback, so perhaps I’m missing something when the authors state, “the feelings are not about the feedback.” Why provide the feedback then? I still struggle with the logic here. Did the bias feedback not affect univalent affect? Is emotional ambivalence not computed as a function of positive and negative univalent affect?

It’s really a confusing causal argument, especially given the order of events. Participants take an IAT, get bias feedback (or not), and then complete measures of affect, defensiveness, and bias awareness. The authors are claiming a (moderated) causal relationship between bias feedback and defensiveness, but not a causal a relationship between bias feedback and affect? And then claim that that whatever affect participants experience is “incidental” even though the measure of affect followed the bias feedback? It’s just hard to wrap my head around.

It’s also important to clarify that positive and negative affect alone aren’t driving the defensive responding or bias awareness, and I appreciate the authors’ new analyses that clarify that ambivalence is doing work above and beyond univalent affect.

That said, I still struggle with what those analyses are doing; they are not straightforward and do not give a full view of the pattern of data. The most defensive people, according to Figure 1, are those who received bias feedback and who were the least ambivalent. And it doesn’t really look like the ambivalent people are more open. It just looks like they’re no more defensive than people who weren’t provided IAT feedback. So can it be said that emotional ambivalence opens people up and makes them less defensive?

My summary assessment is that the logic is a bit strange, the design still puzzles me, and the analyses don’t appear to provide a full, thorough window into what the data actually show. I’ve read and re-read it several times and still have a hard time wrapping my head around it.

7. PLOS authors have the option to publish the peer review history of their article (what does this mean?). If published, this will include your full peer review and any attached files.

Reviewer #1: No

---

## [Author Response · Author response to Decision Letter 1]

1 Feb 2022

Please see Response to Reviewer Document

---

## [Editor Report · Decision Letter 2]

14 Feb 2022

Internal Conflict and Prejudice-Regulation: Emotional Ambivalence Buffers Against Defensive Responding to Implicit Bias Feedback

PONE-D-21-23604R2

Dear Dr. Rothman,

We’re pleased to inform you that your manuscript has been judged scientifically suitable for publication and will be formally accepted for publication once it meets all outstanding technical requirements.

Kind regards,

Natalie J. Shook

Academic Editor

PLOS ONE
---

## [Editor Report · Acceptance letter]

9 Mar 2022

PONE-D-21-23604R2 

Internal Conflict and Prejudice-Regulation: Emotional Ambivalence Buffers Against Defensive Responding to Implicit Bias Feedback 

Dear Dr. Rothman:

I'm pleased to inform you that your manuscript has been deemed suitable for publication in PLOS ONE. Congratulations! Your manuscript is now with our production department. 

Kind regards, 

on behalf of

Dr. Natalie J. Shook 

Academic Editor

PLOS ONE